# Domain Adaptation-based Augmentation for Weakly Supervised Nuclei Detection

N. Brieu[1], A. Meier[1], A. Kapil[1], R. Schoenmeyer[1],
C.G. Gavriel[2], P.D. Caie[2], G. Schmidt[1]

[1] Definiens, Munich, Germany
[2] School Of Medicine, University of St Andrews, UK

**Abstract.** The detection of nuclei is one of the most fundamental components of computational pathology. Current state-of-the-art methods are based on deep learning, with the prerequisite that extensive labeled datasets are available. The increasing number of patient cohorts to be analyzed, the diversity of tissue stains and indications, as well as the cost of dataset labeling motivates the development of novel methods to reduce labeling effort across domains. We introduce in this work a weakly supervised 'inter-domain' approach that (i) performs stain normalization and unpaired image-to-image translation to transform labeled images on a source domain to synthetic labeled images on an unlabeled target domain and (ii) uses the resulting synthetic labeled images to train a detection network on the target domain. Extensive experiments show the superiority of the proposed approach against the state-of-the-art 'intra-domain' detection based on fully-supervised learning.

**Keywords:** Digital Pathology · Cell Detection · Stain Color Normalization · Domain Translation

## 1 Introduction

The analysis of histopathology slides is fundamental to enable a precise and repeatable quantification of cancerous tissue. Some specific applications include the automation of otherwise manual diagnostic scoring methods of e.g. HER2 [15] and PD-L1 [7] stained tissue samples as well as the discovery of novel tissue-based biomarkers [2]. While some analysis solely rely on region segmentation[7], a key prerequisite of most solutions is an accurate nuclei detection. Recent deep learning approaches for nuclei detection and segmentation [5, 14, 10, 9] achieve state-of-the-art performance but demand extensive datasets of manually annotated nuclei centers and manually delineated nuclei boundaries respectively. Because generating manually labeled datasets for nuclei segmentation demands significantly more effort than for nuclei detection and that most applications of quantitative pathology rely more on nuclei detection than on nuclei segmentation, we focus in this work on the sole problem of nuclei detection.

The high number of different cancer indications (e.g. in lung, head and neck, bladder, breast), the vast availability of tissue stains (e.g. HE, HER2, PD-L1) as

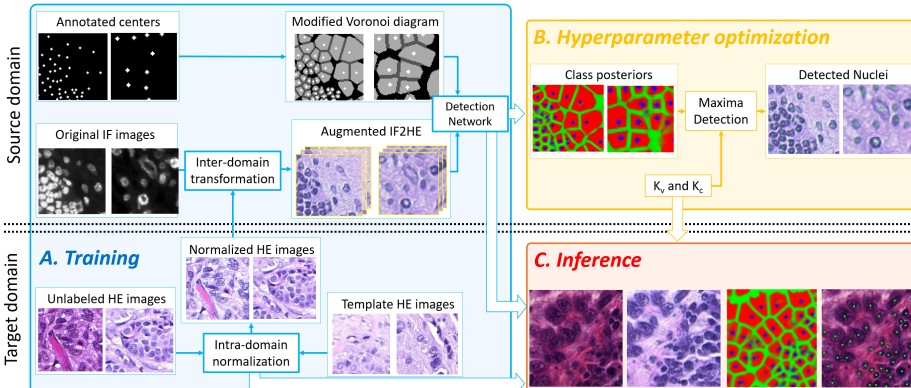

**Fig. 1.** Proposed method for the weakly supervised detection of nuclei on a target domain (e.g. HE) based on labeled images from a different source domain (e.g. IF).

well as the high variability between samples motivates the development of image analysis methods working and reusing information across different domains. This is particularly true for the detection of objects (e.g. nuclei) and regions (e.g. epithelium) which keep a relative morphological consistency across domains. The variability across domains is typically reduced using (i) stain normalization and (ii) domain transformation. Stain normalization methods enforce the visual similarity of images originating from different tissue samples and different patient cohorts but stained with the same tissue stain (e.g. HE) or biomarker (e.g. PD-L1). Recent examples are built on deep convolutional Gaussian mixture model (DCGMM) [16] and unpaired image-to-image translation (CycleGAN) [13]. Domain transformation methods transform images stained with a source stain (e.g. HE) into realistic images synthetically stained with a different target tissue stain (e.g. CD8), using for instance conditional generative adversarial networks (cGANs) [12] or cycle-consistent adversarial networks (CycleGAN) [17].

These recent advances make it possible to leverage labeled images in a first domain to detect objects in an unlabeled second domain. The proposed approach builds on a two-step training methodology [3]: 1) labeled images from a source domain are transformed using unpaired domain adaptation into synthetic versions in a target domain; 2) a convolutional neural network (CNN) is trained on the target domain using the resulting labeled synthetic images. In this standard two-step approach, the images synthesized in the first step are locked in the second step, which hampers an optimal use of the source labeled images. To solve this limitation, we recently introduced the so-called dasGAN network [8] which, by jointly solving the domain adaptation and region segmentation problems, yields a significant improvement of the segmentation accuracy. We present here an alternative approach which more directly builds on the two-step methodology. More precisely, we unlock the full potential of synthetic images by generating for each source image not a single but a series of synthetic images. The detection network is then trained on the resulting augmented ensemble of diverse

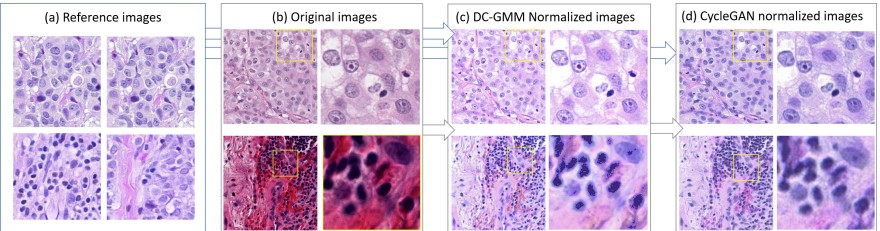

**Fig. 2.** Proposed normalization of HE images. (a) Visually consistent reference images; (b) Input images with high color variability; (c) Results of DC-GMM normalization, resulting in unrealistic patterns in saturated regions; (d) Normalized and realistic looking images obtained with one-to-one domain adaptation between (a) and (c).

but realistic synthetic images. The proposed methodology is weakly supervised: nuclei centers are annotated on the source domain but, because the transformation between the source and the target domain is fully unsupervised, no further annotation is needed on the target domain. In a related approach [6], Hou et al. proposed to generate synthetic nuclear objects as random polygons and to train a generative adversarial network (GAN) for the synthesis of realistic HE images from the resulting masks. Similarly to our approach, this method enables the detection of nuclei in a target stained images without the need for labeled data on this domain. The key benefit of our method is, however, to bypass the complex definition of heuristic rules for the generation of nuclei-like polygons by instead leveraging annotations from another stain domain. Our contribution is twofold: (i) We present the first application of unpaired inter-domain transformation for the weakly supervised detection of nuclei in histopathology images; (ii) We introduce a simple yet accurate approach that improves the standard two-step methodology for domain adaptation and semantic segmentation.

## 2    Methods

As displayed in Fig. 1, our method consists of two main steps: (1) The unsupervised and unpaired transformation of point-labeled source images into synthetic point-labeled target images using CycleGAN; (2) The training of a nuclei detection network based on the synthetic point-labeled images. Because CycleGAN only learns one-to-one domain mapping, the first step further comprises the respective intra-domain normalizations of the source and target stain domains.

### 2.1    Intra-domain Normalization, Inter-domain Translation

While the proposed approach is generic, we present here its application to the transformation between immunofluorescence (IF) and Haemotoxylin and Eosin (HE) stain domains. To fulfill the one-to-one mapping prerequisite, we limit the amount of variability in the respective source and target domains using color-stain normalization. Good normalization is achieved on IF stained images using

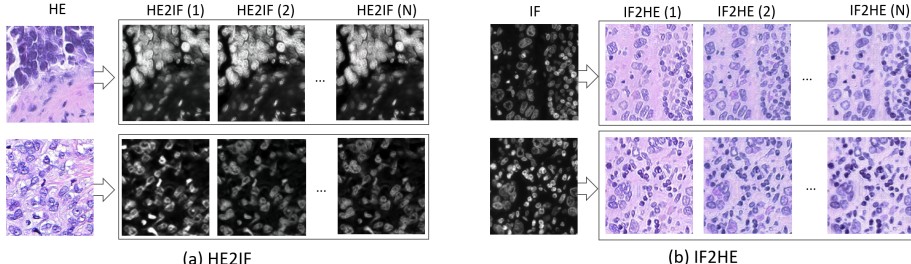

(a) HE2IF                                    (b) IF2HE

**Fig. 3.** Results of unpaired one-to-one 'inter-domain' transformation and augmentation between IF and HE normalized images. (a) HE2IF: HE to IF transformation and augmentation. (b) IF2HE: IF to HE transformation and augmentation.

linear transformation based on minimum and maximum values. For the more complex normalization of HE images, we sequentially combine DCGMM [16] and CycleGAN [13]. The input color variability (b) is decreased using DCGMM, but unrealistic patterns (c) are introduced in over-saturated regions (cf. Fig. 2). This is solved (d) using a one-to-one 'intra-domain' CycleGAN mapping between the DCGMM images and the template HE images. Because visual consistency in the HE and IF domains is enforced, we perform one-to-one 'inter-domain' CycleGAN mapping. Saving the last N training epochs yields an ensemble of translation models with no additional cost nor training complexity. For each labeled image in the source domain, application of these models results in N synthetic images in target domain. These synthetic images are realistic and slightly different in appearance from each other (cf. Fig. 3).

### 2.2   Nuclei Detection

*Voronoi labeling -*   The images in the source domain are labeled with point annotations of nuclei centers. Similar to recent work [11], nuclei detection is formulated as a four-class segmentation problem based on the Voronoi diagram of the annotated centers. Pixels other than the annotated centers are sub-divided into three classes: 1) Voronoi objects, 2) Voronoi edges and 3) background regions. The latter regions are defined as follows. First, for each Voronoi cell $V^i$, we estimate the maximum distance $d^i = max_j [d(c_j, c_i)]$ between the center $c^i$ and the centers $c^j$ of the neighboring cells $V^j$. We then assign the pixels $x \in V^i$ with $d(x, c_i) > d^i$ to the background class. This restricts the Voronoi edge samples to pixels truly located in-between of nuclei. Applying class-based weighting, training is focused on these and the nuclei center pixels. Given the synthetic images and the corresponding Voronoi masks, we train a UNet network with a ResNet18 backbone. Best performing model is selected based on segmentation accuracy on the similarly labeled and domain-transformed validation set.

*Local Maxima Detection -* Nuclei centers are detected as follows: (i) Estimate the Voronoi cells $\hat{V}^i$ by thresholding ($< \kappa_e$) the summed background and edge class-posteriors; (ii) Select center candidates $\hat{c}^i$ as the respective maxima of the center

class-posterior $p_c$ in each cell $\hat{V}^i$; (iii) Reject candidates with $p_c(\hat{c}^i) < \kappa_c$. The values of $\kappa_e$ and $\kappa_c$ are optimized by grid search on the validation set to maximize the pairwise matching between the detected and the annotated centers. As in [1], our Voronoi-based approach implicitly accounts for variability in nuclei sizes and does not rely on a fixed kernel size for local maxima detection. For whole slide analysis, we perform detection sequentially on overlapping tiles. Detection results are saved into a single whole slide output file. Given an individual tile, already detected objects are read from this file prior to the analysis and further detection in the proximity of these already detected objects disabled.

## 3   Results

The impact of stain normalization on region segmentation and nuclei detection is well documented [1, 4]. We report here the first quantitative study on the use of inter-domain transformation for nuclei detection on histopathology images.

### 3.1   Datasets

The IF dataset consists of 75 fields of views (FOV) ($750 \times 750$px) from bladder cancer (MIBC) tissue samples[1] stained with a nuclear IF marker (Hoechst) as well as of 29 FOVs ($400 \times 400$px) from non small cell lung cancer (NSCLC) tissue samples stained with another nuclear IF marker (DAPI). A total of 57K and 15K nuclei centers were manually annotated on the MIBC and NSCLC IF-datasets respectively. The MIBC samples are used for model training and validation, i.e. best model selection and hyper-parameter optimization. The NSCLC samples are used as unseen test set to report detection accuracies on the IF domain, if IF is taken as target domain. The HE dataset consists of 142 FOVs ($740 \times 740$px) selected on NSCLC tissue samples from the TCGA Research Network database and of 30 FOVs ($1000 \times 1000$px) from breast cancer samples from a proprietary dataset. A total of 65K nuclei were annotated on these two datasets, which are further merged and used for training and validation. We use the training set of the TNBC[10] and MoNuSeg[9] datasets, as unseen test sets to report detection accuracies on the HE domain, if HE is taken as target domain.

### 3.2   Experiments and Results

We study two setups for the availability of labeled data on the target domain. First, we assume the target domain to be unlabeled and train the detection network solely on the synthetic images that were generated from the complete set of labeled images in the source domain. Second, an increasing amount of labeled images from the target domain is additionally employed for training. In the first

[1] We thank Ms Frances Rae and the NHS Lothian Tissue Governance Unit for providing the patient samples, Ethical status/approval ref: 10/S1402/33, conforming to protocols approved by East of Scotland Research Ethics Service (REC)

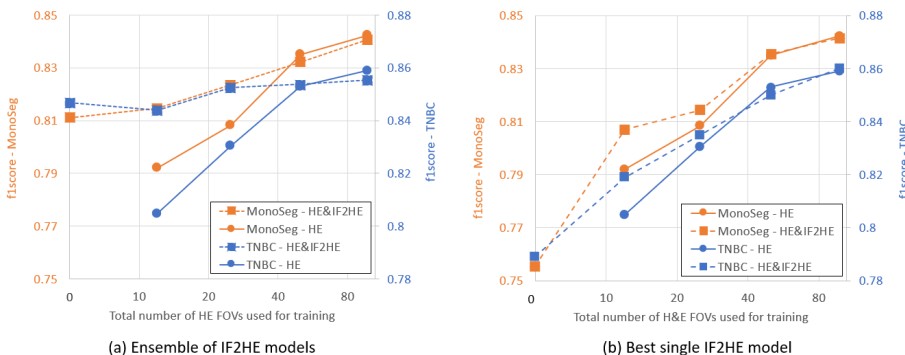

(a) Ensemble of IF2HE models          (b) Best single IF2HE model

**Fig. 4.** Detection accuracy with inter-domain (HE&IF2HE - $N_{HE} = 0$), intra-domain (HE only), and cross-domain (HE&IF2HE - $N_{HE} > 0$) supervisions for increasing availability of labeled nuclei $N_{HE}$ in the target domain (HE), if (a) an ensemble or (b) solely the best of the IF2HE inter-domain transformation models are considered for generation of the synthetic IF2HE images.

setup, both cases of (i) IF as unlabeled target domain and HE as labeled source domain (HE2IF) and of (ii) HE as unlabeled target domain and IF as labeled source domain (IF2HE) are investigated. In the second setup, experiments are focused on the latter and most challenging IF2HE case. Reported detection f1 scores are based on Hungarian matching between annotated and detected centers, with a maximum allowed distance between matched centers of $5\mu m$ .

*Inter-domain supervision* - Out of the last $N = 25$ iterations of the inter-domain CycleGAN training, we select $N = 22$ models based on visual inspection on the transformed IF2HE and HE2IF images. We systematically report the 5-run-average accuracies achieved with the proposed weakly 'inter-domain' supervised approach on the respective unseen test datasets. In the HE2IF case, a f1 score of $f_1 = 0.85$ is achieved on the test NSCLC IF dataset, which is as high as with full supervision based on the complete sets of labeled IF images ($f_1 = 0.85$). In the IF2HE case, f1 scores of ($f_1 = 0.85/0.81$) are obtained on the test TNBC and MoNuSeg HE datasets respectively. While lower, these values are in the same range as with full supervision based on the complete set of labeled HE images ($f_1 = 0.86/0.84$). This shows the ability of the proposed method to detect nuclei using weak inter-domain supervision only.

*Cross-domain supervision* - Fig. 4(a) reports detection accuracies on the target HE domain in case of inter-domain, intra-domain, and cross-domain supervision. For inter-domain supervision, only the labeled images synthetized from the source stain (IF) are employed for training, model selection and hyper-parameter optimization. For intra-domain supervision, only the labeled images in the target domain (HE) are used. For cross-domain supervision, both the labeled images in the target domain (HE) and the labeled images synthetized from the source

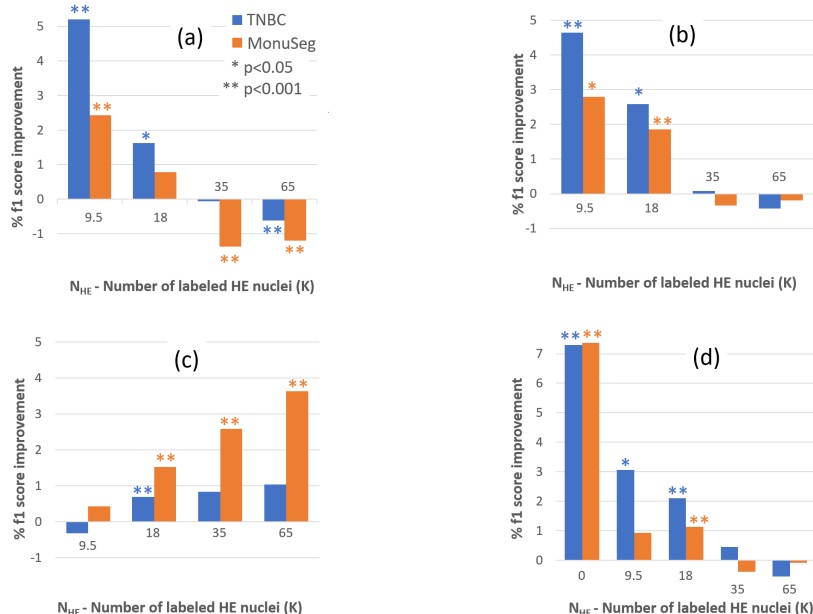

**Fig. 5.** Comparison analysis between different methodologies for domain adaptation and detection: (a) Inter-domain vs. Intra-domain; (b) Cross-domain vs. Intra-domain; (c) Cross-domain vs. Inter-domain; (d) Ensemble vs. Best single CycleGAN models. More precisely, relative improvement of detection accuracy yielded by the first method vs. the second method, and significance testing as measured by paired Student t-Test.

domain (IF) are used in conjunction. All accuracy values are computed on the two unseen MoNuSeg and TNBC datasets. We make the following observations based on Fig. 5: (a) More accurate detection results are obtained with weak inter-domain supervision than with full intra-domain supervision in case of annotation scarcity in the target domain, i.e. if $N_{HE} \leq 18K$; (b) Cross-domain supervision outperforms intra-domain supervision for $N_{HE} \leq 18K$ and reaches similar accuracy level as intra-domain supervision if more HE-stained nuclei are used for training; (c) While being comparable for $N_{HE} < 18K$, cross-domain supervision outperforms inter-domain supervision if more HE-stained nuclei are used; (d) We select the transformation model yielding the highest detection accuracy under inter-domain supervision. Using only this model for generating the synthetic images as in the standard two-step approach, results in a significant drop in detection accuracy. In this case, cross-supervision only results in a marginal improvement compared to intra-domain supervision (cf. Fig. 4(b)).

## 4   Discussion and Conclusion

In this paper, we have presented a novel approach for 'inter-domain' cell detection on a target domain for which no annotation is available, given only cell

center annotations on another source domain. This method builds on recent advances on many-to-one stain normalization and unpaired one-to-one domain transfer for generating series of real but synthetic labeled target images from labeled source images. We have also introduced a 'cross-domain' cell detection method that leverages both synthetic and real labeled target images. Extensive experiments have shown the superiority of the two proposed approaches against the state-of-the-art fully supervised and 'intra-domain' method, based solely on labeled target images. For the near future, we aim to extend this study to images stained with chromogenic immunohistochemistry .

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
