# OpenReview forum: "Domain Adaptation-based Augmentation for Weakly Supervised Nuclei Detection"
_MICCAI.org/2019/Workshop/COMPAY — COMPAY 2019_

### Official Review · AnonReviewer1 · 2019-07-26
**Review for paper #10**

**Rating:** 6
**Confidence:** 3

**Review:**

Summary:
This paper proposes an approach for the extraction of nuclei centers in images images using domain adaptation between source and target domains. Intra- Inter- and cross-domain adaptations are considered.

Strong Points:
- The paper has extensive experiments
- The motivation for using the proposed approach is comprehensible and seems adequate

Constructive criticism:
- If the motivation is well introduced, it is not necessarily well assessed and the practical use is not well detailed.
  Given what I understand, the aim is to detect nuclei in images without using a large dataset of labeled nuclei.
- In Section 2.1, please provide more details on what you mean by "the one-to-one mapping prerequesite"
- Some acronyms (HE, IF) should be detailed for more clarity
- The terms weakly supervised and fully supervised need to be more clearly defined in particular regarding the terms inter-domain
	intra-domain and cross-domain. Given what is said, it seems that cross-domain corresponds to full supervision.
	All these terms should be better defined since this does not ease the understanding of the paper and its conclusion.
- Use the same symbols for the curves in Figures 4(a) and (b) : e.g., TNBC HE&IF2HE is a diamond in (a) and a square in (b).
- The generation of several synthetic images is performed from the last configurations of the trained network.
  However a selection is said to have been done manually (22 models out of the 25 last iterations). Given the results
  it appears that the more the number of generated images, the better the results, so what is the aim of this manual selection
  and is it really helpful ?
- It is not clear why so different results are obtained with TNBC - HE and
- The problem HE2IF seems relatively simple to solve and would benefit from a comparison with the simple computation of a optical density image
- To me the results show that:
	* It is clearly interesting to generate several images if few labels from the target domain are available
	* When many labels from the target domain are used, the ones of the source domain seem not to be very helpful

	So the approach is interesting for the given configuration that few labels are available in the target domain.
	This seems to be what was claimed at the begining of the paper, but the presentation is not enough good to fully understand
	the conclusions.

To conclude the paper is interesting and has some merits but it is difficult to read with many terms that are not clearly defined and this lowers its proper understanding.

---

### Official Review · AnonReviewer3 · 2019-08-08
**Using GANs to adapt to new domains for nuclei segmentation**

**Rating:** 7
**Confidence:** 3

**Review:**

SUMMARY OF MAIN FINDINGS
Deep learning algorithms make up the majority of the state of the art nuclei detection schemes. While deep learning does outperform many methods, deep learning requires a large amount of annotated/ labeled data.
Generative Adversarial Networks (GANs), and its variants, have demonstrated success in digital pathology by effectively mapping the color of a reference distribution to a target distribution. In this paper the authors propose a two-step unsupervised method to detect nuclei in a dataset that does not have any annotations/labels. Source domain (IF Images) will be used in reference to a dataset that has associated labels, while target domain (H&E Images) references a dataset without associated labels and is the desired dataset for detection.

First, the source domain dataset is transformed to the target domain using a proposed variant of the GANs architecture, dasGAN. A CNN segmentation scheme is then trained on the transformed domain set that has available annotations. The developed model is then used to detect nuclei on the target domain set. This experiment is repeated by interchanging the transformations, where the H&E set is then used as the source domain and IF is used as the target domain. A third experiment is conducted, cross-domain supervision, where synthesized images and original images are used for training.
The authors report that models trained on synthesized images and labels outperformed models that were trained on original images and labels only. However, cross-domain trained models produced better accuracies than the latter, and reached similar accuracies to models trained on a large number of original images and labels.

DETALED COMMENTS
The strengths include:
1. The authors propose a novel and clever method for creating labels/annotations for datasets that originally do not have any. They address an important problem in deep learning/ digital pathology and demonstrate positive results.

---

### Official Review · AnonReviewer2 · 2019-08-12

**Rating:** 8
**Confidence:** 4

**Review:**

The authors present a method to convert from fluorescence images to H&E and vice-versa via cycleGANs. These converted images are subsequently used to train nuclei detection algorithms without needing to have detailed annotations in both domains, but only in one domain. I think this is a highly relevant area and the results are convincing. The paper includes extensive experiments including learning curves for different data set sizes. I do have some comments:

- p-values are mentioned in Figure 5, but there is no description of the statistical test used to obtain these p-values, this should be added.
- some qualitative results on the nuclei detection task would have been nice as well.

---

### Decision · Program_Chairs · 2019-08-20

Accept